# Investigation of the Manufacturing Orientation Impact on the Mechanical Properties of Composite Fiber-Reinforced Polymer Elements in the Fused Filament Fabrication Process

**DOI:** 10.3390/polym15132757

**Published:** 2023-06-21

**Authors:** Emil Spišák, Ema Nováková-Marcinčínová, Ľudmila Nováková-Marcinčínová, Janka Majerníková, Peter Mulidrán

**Affiliations:** 1Faculty of Mechanical Engineering, Technical University of Košice, Letná 9, 042 00 Košice, Slovakia; ema.novakova-marcincinova@tuke.sk (E.N.-M.); janka.majernikova@tuke.sk (J.M.); peter.mulidran@tuke.sk (P.M.); 2Faculty of Manufacturing Technologies with a Seat in Prešov, Technical University of Kosice, Bayerova 1, 080 01 Prešov, Slovakia; ludmila.novakovamarcincinova@tuke.sk

**Keywords:** rapid prototyping, 3D printing, FDM, FFF, composite, extruder, filament, PLA plastic, carbon, bronze

## Abstract

This article examines the mechanical properties and compatibility of selected composite materials produced with RP technology and the FFF—fused filament fabrication process. The article scales sophisticated modern materials based on PLA—polylactic acid—plastic and its composite variants. The research is carried out on the 3D FFF printer Felix 3.1 with a dual extruder, which works on the “open-source” principle. In this research, elements of the paradigm and methodology of the processing technology for RP were applied; they were implemented according to EN ISO 527 and ISO 2602 standards. The aim of this study was to investigate the impact of 3D-printing strategy on the mechanical properties of 5 types of PLA composites. The results of this research solve the material compatibility problem, primarily through experimental testing of different combinations of filaments in different printing directions. Analysis of the experimental data showed correlations between the choice of printing strategy and mechanical properties, mainly tensile strength of the selected filaments. The research results show the influence of the printing orientation on mechanical properties of 3D printed samples: parts extruded in length orientation showed higher values of tensile strength compared to parts made in width and height. The CarbonPLA material exhibited 10 times higher tensile strength when printed in length compared to samples.

## 1. Introduction

The fused deposition modeling (FDM) method is one of the most widespread methods of 3D printing by melting plastic material and was patented in 1989 by Steven Scott Crump. Steven is also behind the designers of Stratasys, one of the leading companies producing FDM printers [1].

In recent years, 3D printing technology has gained significant attention in various industries due to its ability to fabricate complex structures with precision and efficiency. One area of interest is the development of reinforced polymers, where the addition of fillers or fibers enhances the mechanical properties of the printed parts. Among these materials, reinforced polylactic acid (PLA) has emerged as a promising candidate, offering a combination of biocompatibility, ease of processing, and good mechanical performance [2,3,4].

The widespread adoption of 3D printing technology has paved the way for innovative advancements in material manufacturing. Among the various materials utilized in 3D printing, reinforced polylactic acid (PLA) composites have garnered significant interest due to their enhanced mechanical properties and broad applicability. The combination of PLA, a biodegradable and easily processable polymer, with reinforcing agents such as fibers or fillers, offers the potential for producing durable and functional components [5,6,7].

Some studies deal with similar research, such as Hsueh et al. [8]. Their work characterized the polylactic acid (PLA) and polyethylene terephthalate glycol (PETG) materials of FDM under four loading conditions (tension, compression, bending, and thermal deformation) in order to obtain data regarding different printing temperatures and speeds. Jiang et al. [9] published a paper presenting a study of CFF FFF parts produced on desktop 3D printers using commercially available filament. Tensile test samples fabricated with CFF polymer composite and unfilled polymer were printed and then tested following ASTM D3039M. Test bars were printed with FFF bead orientations aligned with the direction of the applied load at 0°, 45°, ±45°, and normal to the loading axis at 90°. Results for tensile strength and tensile modulus show that CFF coupons in general yield higher tensile modulus at all print orientations and higher tensile strength at 0° print orientation. Casavola et al. [10] described the mechanical behavior of FDM parts through classical laminate theory and measured the elastic modulus values in the longitudinal and transverse fiber directions. Lee et al. [11] concluded that screen angle, air gap, and layer thickness affect the elastic performance of flexible ABS objects. Also, Anitha et al. [12] studied the effect of layer thickness and showed that the performance increases as the thickness decreases. Ahn et al. [13] conducted several experiments to determine the effects of air gap, bead width, grid orientation, and ABS color on tensile and compressive strength. They determined that the air gap and grid orientation have a significant effect on the tensile strength; on the other hand, the other parameters have a negligible effect. Furthermore, the factors they investigated do not significantly affect the compressive strength. Sood et al. used response surface methodology to analyze the functional relationship between sample strength and several factors, e.g., orientation of the structure, layer thickness, grid angle, and air gap. Their results show that these factors affect bonding and deformation within parts. Saharudin et al.’s [14] work presents the results of mechanical tests of models manufactured with two 3D printing technologies, FDM and CFF. Both technologies use PLA- and PA-based materials reinforced with carbon fibers. The work includes both uniaxial tensile tests of the tested materials and metrological measurements of surfaces produced with two 3D printing technologies. The test results showed a significant influence of the type of technology on the strength of the models built and on the quality of the technological surface layer. After the analysis of the parameters of the primary profile, roughness, and waviness, it can be clearly stated that the quality of the technological surface layer is much better for the models made with the CFF technology compared to the FDM technology. The works of Wysmulski et al. [15] and Rozylo et al. [16] were focused on studying the mechanical properties of 3D printed thin wall samples. They used numerical simulation to predict these mechanical properties. Numerical results were compared with the experimental ones.

Conversely, few articles in the literature deal with the development of predictive models to determine the mechanical properties of FDM parts [17,18,19]. To establish a strong foundation for this study, an extensive review of relevant research articles has been conducted. To fully exploit the potential of reinforced PLA in 3D printing applications, it is crucial to understand the influence of various printing strategies on the resulting mechanical properties. This article aims to provide a comprehensive examination of the 3D printing strategies employed for 5 types of reinforced PLA and their impact on the mechanical properties of the printed components.

## 2. Materials and Methods

### 2.1. Material Filaments Used for Experimental Research

The CarbonFil fiber is based on a unique HDglass blend of PETG. It is reinforced with 20% ultralight and relatively long carbon fibers, resulting in an exceptionally stiff carbon-reinforced fiber. CarbonFil is twice as stiff as HD glass and yet is 10% more impact resistant, a remarkable feature for carbon-reinforced fiber [20].

Metal File Bronze is a PLA-based metal-filled fiber with approximately 80% gravimetric bronze filling. This incredibly high fill of bronze powders allows any FDM 3D printer user to print bronze objects indistinguishable from original bronze castings. MetalFil printed objects are very easy to process, making it possible to create stunning bronze objects with various patina effects, including a real bronze look and “cold” bronze metal. It is approximately 300% heavier than PLA (density 3.5 g/cm^3^). It is easy to process. MetalFil material can be brushed, sanded, polished, waxed, and painted after processing with a patina. It is very easy to print on a direct drive, and the use of Bowden extruders, cold printing, and lack of deformation after cooling are truly unique features for metal-filled filament. It has improved flow behavior and interlayer adhesion [20].

The basic material used in experimental printing is PLA. It is a biodegradable plastic due to its natural origin (corn, sugar cane, or potatoes). By adding other materials to PLA, it is possible to obtain filaments such as wood (particles of wood), plaster (plaster), and BronzeFill (bronze). Conductive PLA is also produced and is designed for small voltages and currents. It is more flexible than classic PLA but at the cost of less adhesion between layers. If the printing temperature is lower than 225 °C, the resulting surface is glossy, but if the temperature is higher than 225–230 °C, the surface will be matte [21,22,23].

### 2.2. Materials Compatibility

The problem of material compatibility is primarily about experimental testing of different combinations. Some materials are difficult and unrealistic to print together because they do not bond together. It requires experimental testing and setting parameters such as extruder or substrate temperatures, print speed, nozzle-to-substrate distance, and more. Individual materials are generally divided into three groups: compatible, experimentally compatible, and incompatible combinations. If we can print materials close to the extruder temperature, those materials will stick together. The subject of our research was to determine which of the investigated composites is the strongest one.

### 2.3. Preparation and Printing of Tested Composite Samples

This experimental research aims to perform tensile tests based on theoretical knowledge and verify and compare the results. Composite printing using two extruders gives us great diversity in terms of its use and arrangement. However, not all variants that may seem practical at first glance are suitable in practice. Therefore, it is necessary to verify the various material designs through 3D printing of samples and analysis of different materials by measuring, for example, the tensile strength and elongation with a tensile test machine. According to the required application, it is possible further to observe dimensional stability, surface treatment, or other properties [5,6,7,8]. This research uses a 3D FFF printer Felix 3.1 with a dual extruder (Figure 1) for producing test samples. The software for controlling the printer is Repetier-Host V.2.0.5, which then processes the G code according to any slicer. In this case, Slic3r Prusa Edition version 1.35 was used. The CAD software Creo is used for modelling individual parts.

The test samples that were produced on the printer mentioned above correspond to the EN ISO 527 standard [24]. In this case, 30 test specimens were produced from each composite (Figure 2). According to ISO 2602, accuracy and probability are determined by a 95% confidence interval [25,26].

In our case, the temperature for the working nozzle of the extruder was set to 220 °C and 60 °C was used for the working plate. Subsequently, the composite test samples were extruded from the extruders layer by layer, alternately from two material filaments. This was continued until the desired shape of the composite sample was created.

Composite test specimens extruded to length by an extruder are formed from 52 layers with a total length of 150 mm and a width of 20 mm. Composite test samples extruded by an extruder in width are made of 10 layers with a total thickness of 4 mm. Test samples produced in height are made of 378 layers with a total height of 150 mm.

As part of this experimental research, 150 test samples were printed. Thirty samples for each tested material were printed, from which 10 composite samples were printed in length, 10 composite samples in width, and 10 composite samples in height.

### 2.4. Experimental Tensile Test of Composite Materials and Its Evaluation According to EN ISO 527 and 2602

When evaluating the results of the static tensile test for the calculation of the tensile strength, measurement results were evaluated, which served to determine the result of the most probable values of the measured quantities and the size of the errors. Part of the evaluation is also the analytical approximation of the acquired dependencies using appropriate mathematical and statistical methods. Following the standard Plastics—Determination of tensile properties EN ISO 527-1, which in the first part describes the general principles, the accuracy of the measured values is determined by calculation using a confidence interval with a probability of 95% according to EN ISO standard no. 2602—including statistical interpretation of test results and mean estimation and named confidence interval. One of the basic standards developed by the ISO/TC 69 technical committee is the EN ISO 2602 standard, which focuses on data’s statistical interpretation. The purpose of these standards is to provide a faithful interpretation of the statistical calculations of the recorded test data and thereby create the prerequisites for the correct use of the results obtained for testing and certification in various industrial applications of statistical methods and in the application of other ISO and IEC standards that can use these methods. This standard specifies a precise procedure for obtaining an estimate of the mean and calculating a two-sided and a one-sided confidence interval for the mean using the standard deviation estimate or the sampling interval (Table 1) [26].
(1)$$s_{1}\;=\;\sqrt{\frac{1}{n\;-\;1}\;\sum_{i\;=\;1}^{n}{\left(\sigma_{M1i}\;-\;\bar{\sigma_{M1}}\right)}^{2}}$$
(2)$$\bar{\sigma_{M1}}-\frac{t_{0.975}}{\sqrt{n}}\cdot s_{1}<m<\bar{\sigma_{M1}}+\frac{t_{0.975}}{\sqrt{n}}\cdot s_{1}$$
(3)$$m\;<\;\bar{\sigma_{M1}}\;+\;\frac{t_{0.95}}{\sqrt{n}}.\;\;s_{1}\;\;m\;>\;\bar{\sigma_{M1}}\;-\;\frac{t_{0.95}}{\sqrt{n}}.\;\;s_{1}$$

To determine the mechanical properties of test composite samples from filament materials, tensile tests were performed. Tensile tests were conducted on VEB TIW TIRAtest 2300 testing machine (TIRA Maschinenbau GmbH, Rauenstein, Germany) shown in Figure 3.

## 3. Results

### 3.1. Tensile Test Results

The measured and calculated values obtained during the implementation of the experiments are listed for individual composite test samples in the following table (Table 2). Measured force values F_m_ represent forces at the tensile strength. From the given force values, it is possible to determine the tensile strength limit σ_M_ and the proportional elongation at tensile strength ε_M_ (Table 2). Test samples after tensile test fracture are shown in Figure 4a. In Figure 4b are CarbonPLA samples after the tensile test.

Figure 5, Figure 6, Figure 7, Figure 8, Figure 9 and Figure 10 show the dependence of force on displacement for 5 types of reinforced 3D printed samples. Each figure represents 3 types of printing strategies of one type of printed material.

Figure 5, Figure 6, Figure 7, Figure 8, Figure 9 and Figure 10 show the dependence of the displacement on the loading force F, which was evaluated using the data from the testing machine software during the static tensile test.

### 3.2. Documentation of Microstructures of Selected Composite Samples

The samples were observed at the Technical University in Košice in the laboratory of light microscopy and metallography under constant ambient conditions using a Keyence VHX—500 digital microscope (Keyence SV, Mechelen, Belgium). This microscope has a resolution of 18 million pixels. It is shown in Figure 11.

When observing the microscopic structure of a sample made of Carbon PLA filament along its length, one can see the failure of the fibers due to the plastic fracture to the narrowing (Figure 12a), as well as the brittle fracture at the point of attachment of the individual layer of the filament. It is also possible to observe the filling of the samples and the interconnectedness of the individual layers when printing the samples in width (Figure 12b).

Microscopic observation of the fracture, after the tensile strength test was performed on the sample of PLA Carbon + PLA bronze in width, shows the fracture of the printed filaments that occurred during this test. When observing the fracture, the PLA CarbonPLA filament breaks into a taper after stretching, and the PLA Bronze filament has a visibly brittle fracture. From the microscopic photo, it is also clear that the individual layers of filaments are bonded, when alternating layers of Carbon filament and Bronze filament are printed (Figure 13a). In Figure 13b it is also possible to observe the chosen route for the filling when printing samples from PLA Carbon + PLA Bronze filament in portrait.

From observation of the PLA Carbon + Bronze sample in width (Figure 14) under the microscope, it is obvious that during the tensile test this sample suffered a fracture. On the end parts of some of the printed filaments, fracture can be observed until the narrowing caused by pulling of PLA Carbon filament.

### 3.3. Evaluation of Composite Tested Samples with an Emphasis on Their Production Orientation

The results of the tensile strength measurements were calculated in MPa and recorded in a graphic diagram, and at the same time evaluated according to individual composite materials and the orientation of the produced composite test sample in length, height, and width (Figure 15, Figure 16 and Figure 17).

A graphic evaluation of the strength of the PLA Carbon composite material proves that the strongest are the samples printed in length, which show the largest strength limit, namely 21.2 MPa.

Graphical evaluation of the strength of the composite material PLA Carbon in combination with pure PLA plastic white proves that the samples printed in length show the highest limit of strength which is 25.8 Mpa and thus demonstrate higher strength than the samples printed in height.

Graphical evaluation of the strength of the bronze composite material shows that, in this case, the samples printed in width show the highest strength limit, namely 20.2 MPa, they are stronger compared to the samples printed in height. Based on the results from the graph, it can be concluded that the composite samples produced in height have the lowest strength limit, and the samples produced in length have the highest strength limit. Only in the case of the composite material Bronze is the strength limit of the produced samples higher when printed in width.

Graphical evaluation of the composite material bronze in combination with composite carbon fiber shows the most significant strength when printed in length, namely 22.7 MPa.

Graphical evaluation of the strength of the composite material Bronze in combination with PLA plastic white printed lengthwise shows a strength limit of 25.1 MPa.

Based on a comprehensive graphic evaluation of the comparison of individual composite materials, when printed lengthwise, samples made of carbon composite material in combination with white PLA plastic demonstrate the highest strength with a value of 25.8 MPa.

An evaluation of the comparison of individual composite materials, when printed in height, shows the highest strength in samples of bronze composite material with a strength of 2.7 MPa. A comparison of individual composite materials when printing in landscape shows that with this method of printing, the carbon material in combination with white PLA plastic shows the highest strength limit, namely 21 MPa.

As a result of changing the strategy of printing filaments in three directions, the highest tensile strength of the test sample was recorded at 25.8 MPa in the combination of the material Carbon PLA and PLA plastic white, compared to the composite Bronze and PLA plastic white printed lengthwise and compared to samples printed widthwise. At the same time, it was proven that samples printed in height showed the lowest strength compared to samples printed in length and width.

## 4. Discussion

Previously published research on the process of 3d printing of reinforced PLA materials mainly focused on the investigation of material and process parameters and their impact on the mechanical properties and surface parameters [27,28,29]. Evaluation of mechanical properties, mainly tensile strength, of 5 types of reinforced PLA materials was the subject of the presented work. The research also focused on the printing strategy and its impact on the deformation behavior. Other works [15,16,30] were also aimed at predicting mechanical properties, deformation behavior of different types of reinforced 3D printed material using numerical simulation The work of Guessasma et al. [31] involved experiments and numerical simulations of carbon reinforced PLA material with a focus on the infill parameters. They studied the impact of infill rate on the tensile properties.

The experimental results in this work showed the importance of printing strategy on the mechanical properties and deformation behavior of printed samples. The values of tensile strength σ_M_ and the proportional elongation at tensile strength ε_M_ are shown in Table 1. These results suggest that printing strategy has a significant impact on the tensile strength and deformation behavior of tested materials. In addition, the types of filament, in our case, PLA-based filaments with carbon or bronze reinforcement, were also studied, and their impact on mechanical properties was evaluated using uniaxial tensile tests. Moreover, stress-strain behavior of 5 tested filaments was compared for samples made using three types of printing strategy in Figure 5, Figure 6, Figure 7, Figure 8, Figure 9 and Figure 10. The printing strategy has a significant effect on the results of the tensile tests. The printing strategy that produced tensile test samples in height showed lowest values of tensile strength and elongation at break. Thus, based on making these samples, this printing strategy is unsuitable for the production of parts that will experience tensile load. Higher values of tensile strength were measured for samples produced in width compared to samples produced using printing strategy “height”. These values were up to 10 times higher compared to samples produced in height. The highest values of tensile strength were measured when samples were printed in length for the majority of cases. Only Bronze PLA material that was printed in width had higher values of tensile strength compared to samples printed in length. A comparison of all types of filaments produced in length is shown in Figure 10, which shows that Carbon PLA material showed much higher values of elongation compared to other materials.

## 5. Conclusions

Considering the possibility of more massive application in practice of even cheaper rapid prototyping devices to produce final products, it is necessary to conduct research on the possibility of increasing the mechanical properties of products produced on cheaper and more easily available devices for rapid prototyping, as well as devices using fused deposition modeling technology. The results of this paper can help with FFF-RP methods, which offer us many variants of how individual models produce and apply elements of the paradigm and methodology of process technologies for DP and 3D printing. The results from experiments were used by a Czech car manufacturer to optimize the production of 3D printed prototypes of certain car parts and production jigs.

The experimental tests showed that by using carbon composite as a material it is possible to increase the tensile strength of the PLA plastic used, and thus, the mentioned technology can be used in practice even for more demanding applications. The highest strength was achieved with samples made of CarbonPLA + PLA white and Bronze + PLA white composites in the length direction. Strength values for all investigated materials dropped significantly when testing samples were printed perpendicular to the applied force. For the samples produced in this way, the strength reached about 10% of the strength of the samples printed in the direction of the stress force. Experiments have proven that with a suitable printing method we can create prototype production algorithms that correspond to the required explicit application in the production of a model or component. Based on the analyses performed and the results achieved, it can be concluded that:

The use of a suitably chosen composite material as a reinforcing substance can influence and change selected mechanical properties of products produced by the fused deposition modeling method in rapid prototyping technologies compared to the use of PLA plastic material, which is the most often used material in the mentioned technology.In the printing strategy, part orientation is an important parameter that must be considered when printing parts made of reinforced PLA material. Printing strategy should be also based on the type of load that the part can experience in practical use.Most of the reinforced PLA materials tested showed very low values of elongation and plastic deformation. Only the PLA Carbon material printed in length experienced good plasticity.

## Figures and Tables

**Figure 1 polymers-15-02757-f001:**
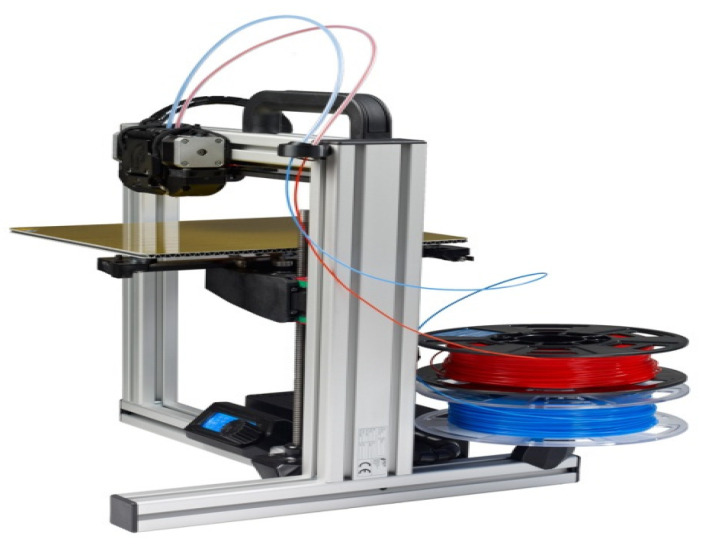
FFF printer Felix 3.1 with dual extruder.

**Figure 2 polymers-15-02757-f002:**
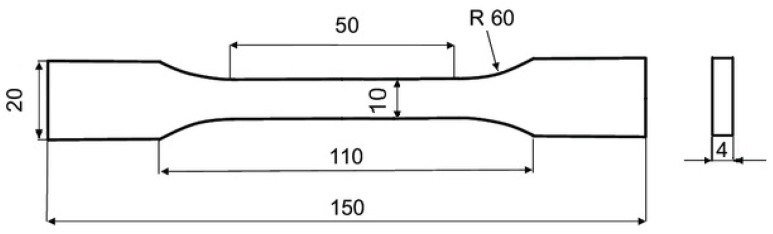
Specimen tensile tests dimensions according to ISO 527 [24].

**Figure 3 polymers-15-02757-f003:**
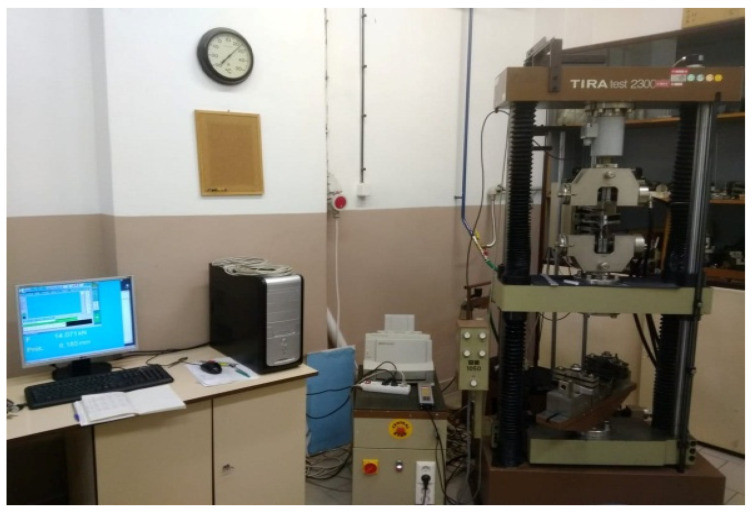
TiraTest 2300 testing machine.

**Figure 4 polymers-15-02757-f004:**
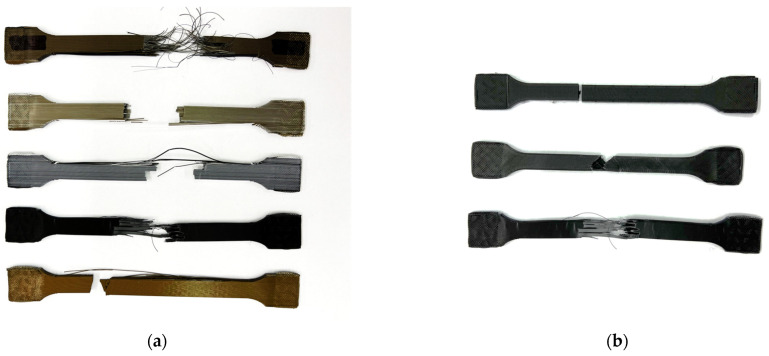
Test samples after tensile test (**a**) in length, (**b**) Carbon PLA in length, width and height.

**Figure 5 polymers-15-02757-f005:**
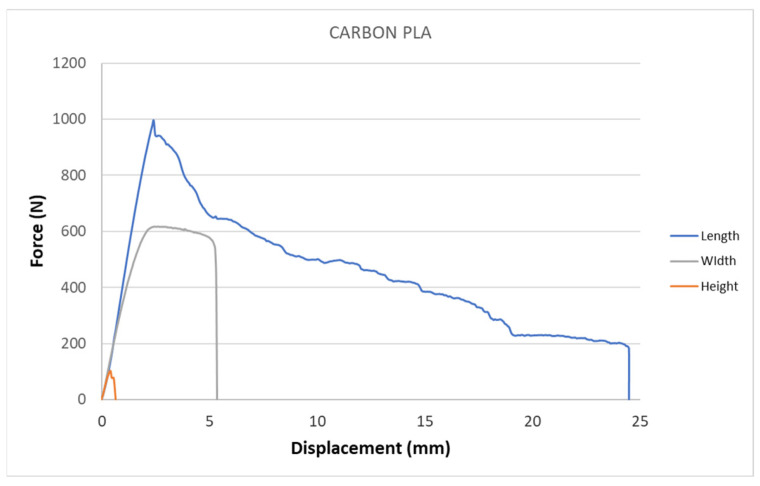
Diagram of tensile test of Carbon PLA composite samples—length, height, width extrusion strategy.

**Figure 6 polymers-15-02757-f006:**
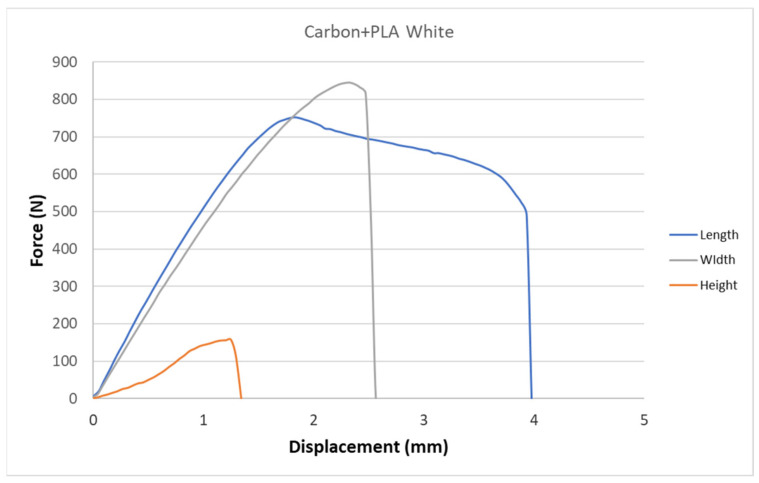
Diagram of tensile test of composite samples Carbon +PLA white—length, height, width extrusion strategy.

**Figure 7 polymers-15-02757-f007:**
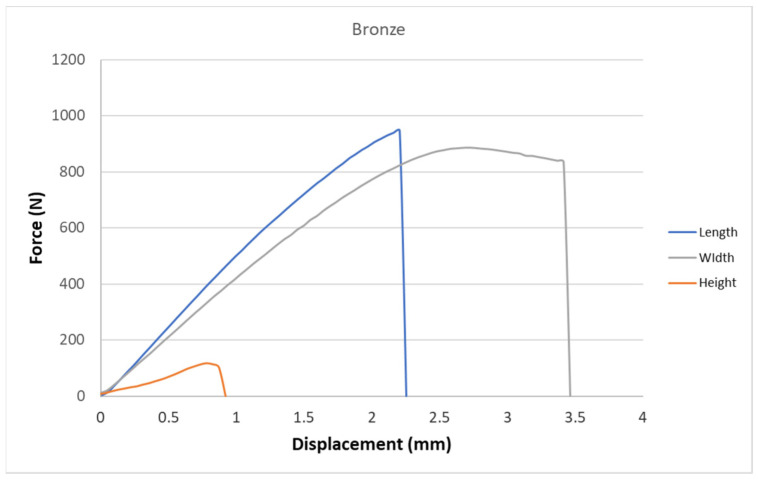
Diagram of tensile test of composite samples Bronze—length, height, width.

**Figure 8 polymers-15-02757-f008:**
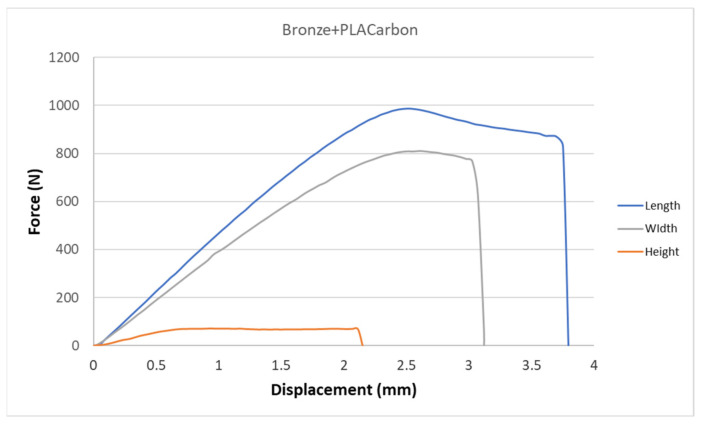
Diagram of tensile test of composite samples Bronze + Carbon PLA—length, height, width extrusion strategy.

**Figure 9 polymers-15-02757-f009:**
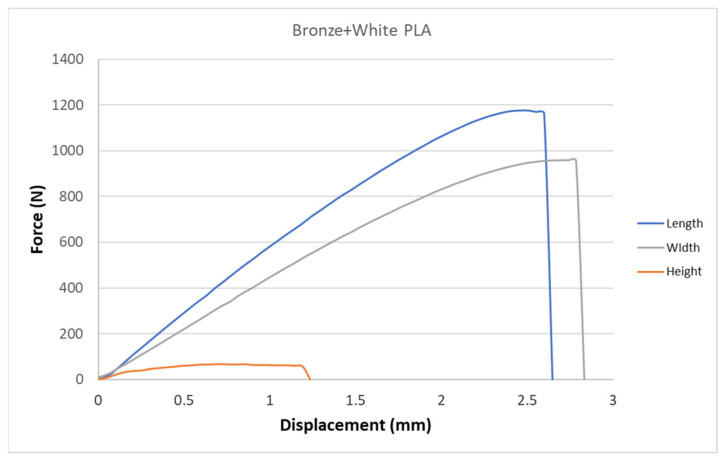
Diagram of tensile test of composite samples Bronze + PLA white—length, height, width extrusion strategy.

**Figure 10 polymers-15-02757-f010:**
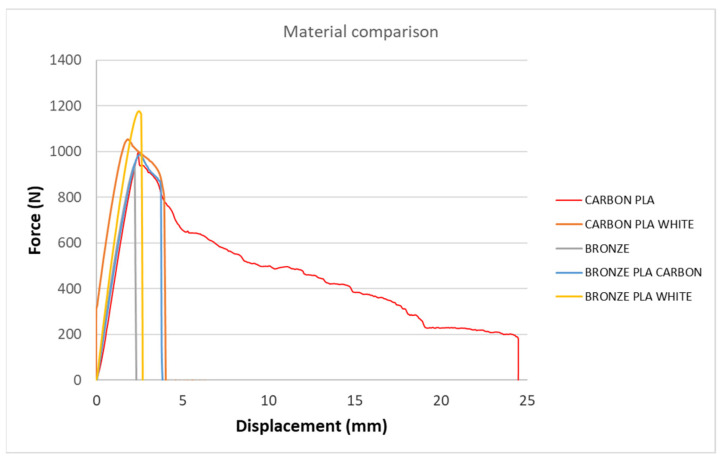
Diagram of tensile test of different types of composite samples—length type of extrusion.

**Figure 11 polymers-15-02757-f011:**
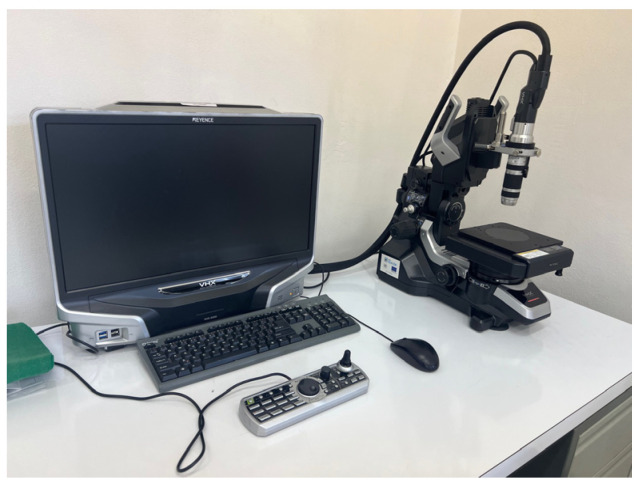
Keyence VHX-500 digital microscope.

**Figure 12 polymers-15-02757-f012:**
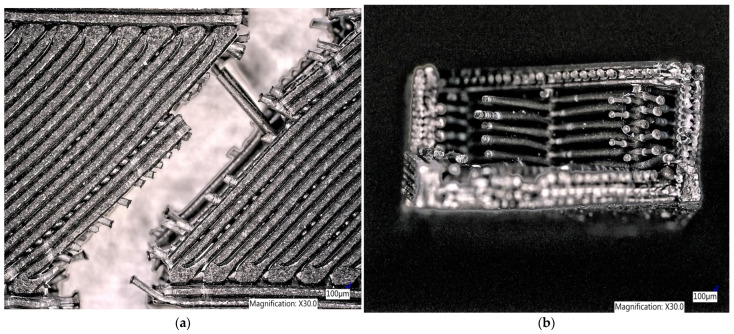
Microscopic structure of a sample made of Carbon PLA filament (**a**) in length (**b**) in width.

**Figure 13 polymers-15-02757-f013:**
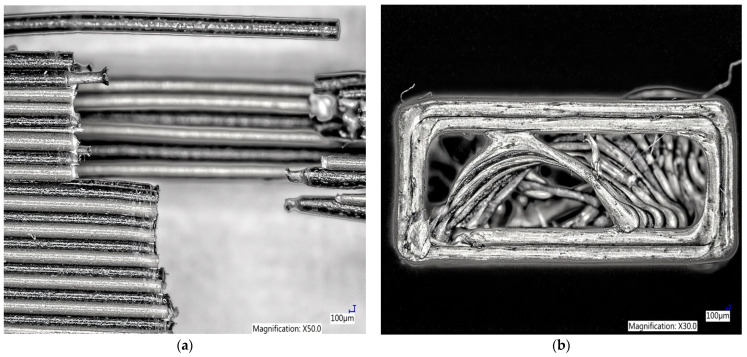
Microscopic structure of a sample made of Carbon PLA + PLA Bronze filament (**a**) in width (**b**) in height.

**Figure 14 polymers-15-02757-f014:**
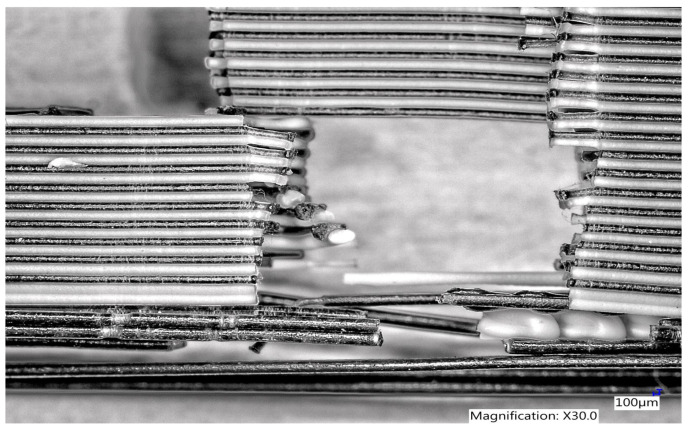
Microscopic structure of a sample made of PLA Carbon + Bronze filament in width.

**Figure 15 polymers-15-02757-f015:**
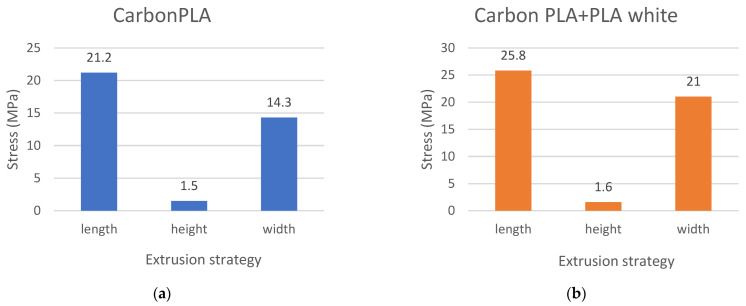
Comparison of (**a**) Carbon PLA and (**b**) CarbonPLA+White samples printed using different printing strategies.

**Figure 16 polymers-15-02757-f016:**
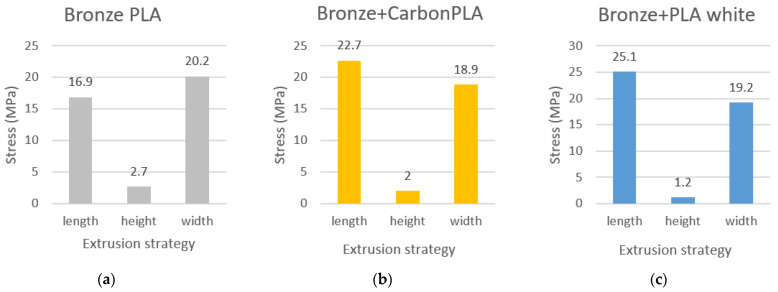
Comparison of Bronze samples made of three types of filaments using different printing strategies (**a**) Bronze PLA, (**b**) Bronze + CarbonPLA and (**c**) Bronze + PLA white.

**Figure 17 polymers-15-02757-f017:**
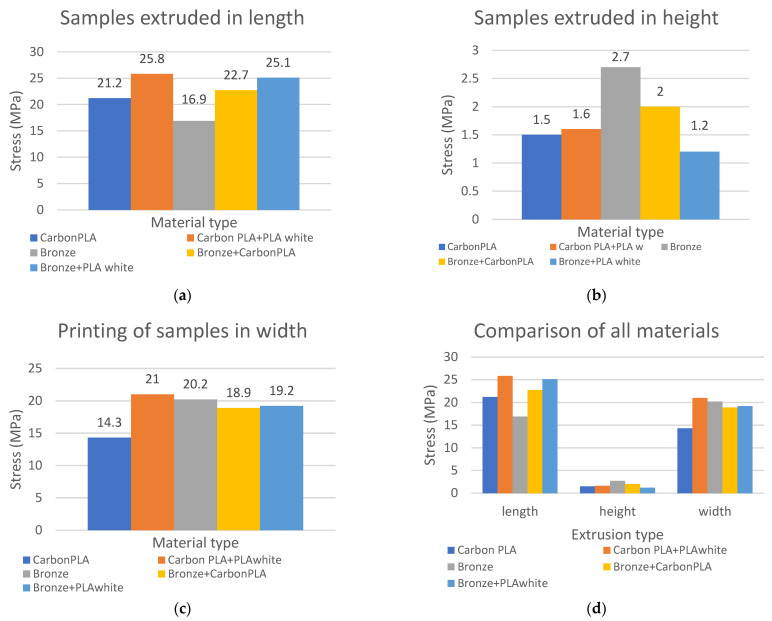
Comparison of tensile strength made of five types of filaments using different printing strategies (**a**) samples extruded in length, (**b**) samples extruded in height, (**c**) samples of samples in width, (**d**) comparison of all materials.

**Table 1 polymers-15-02757-t001:** Coefficient values t1-α and ratio t1-α/$\sqrt{n}$ for the t-distribution s v = n − 1 determination of freedom.

	**Confidence Level Two-Sided Case**		**Confidence Level One-Sided Case**			**Confidence Level Two-Sided Case**		**Confidence Level One-Sided Case**	
	95%	99%	95%	99%		95%	99%	95%	99%
n	t_0.975_	t_0.995_	t_0.95_	t_0.99_	n	$\frac{t0.975}{\surd n}$	$\frac{t0.995}{\surd n}$	$\frac{t0.95}{\surd n}$	$\frac{t0.99}{\surd n}$
2	12.71	63.66	6.314	31.82	2	8.985	45.013	4.465	22.501
3	4.303	9.925	2.920	6.965	3	2.484	5.730	1.686	4.021
4	3.182	5.841	2.353	4.541	4	1.591	2.920	1.177	2.270
5	2.776	4.604	2.132	3.747	5	1.242	2.059	0.953	1.676
6	2.571	4.032	2.015	3.365	6	1.049	1.646	0.823	1.374
7	2.447	3.707	1.943	3.143	7	0.925	1.401	0.734	1.188
8	2.365	3.499	1.895	2.998	8	0.836	1.237	0.670	1.060
9	2.306	3.355	1.860	2.896	9	0.769	1.118	0.620	0.966
10	2.262	3.250	1.833	2.821	10	0.715	1.028	0.580	0.892
11	2.228	3.169	1.812	2.764	11	0.672	0.956	0.546	0.833
12	2.201	3.106	1.796	2.718	12	0.635	0.897	0.518	0.785
13	2.179	3.055	1.782	2.681	13	0.604	0.847	0.494	0.744
14	2.160	3.012	1.771	2.650	14	0.577	0.805	0.473	0.708
15	2.145	2.977	1.761	2.624	15	0.554	0.769	0.455	0.668
16	2.131	2.947	1.753	2.602	16	0.533	0.737	0.438	0.651
17	2.120	2.921	1.746	2.583	17	0.514	0.708	0.423	0.627
18	2.110	2.898	1.740	2.567	18	0.497	0.683	0.410	0.605
19	2.101	2.878	1.734	2.552	19	0.482	0.660	0.398	0.586
20	2.093	2.861	1.729	2.539	20	0.468	0.640	0.387	0.568
21	2.086	2.845	1.725	2.528	21	0.455	0.621	0.376	0.552
22	2.080	2.831	1.721	2.518	22	0.443	0.604	0.367	0.537
23	2.074	2.819	1.717	2.508	23	0.432	0.588	0.358	0.523
24	2.069	2.807	1.714	2.500	24	0.422	0.573	0.350	0.510
25	2.064	2.797	1.711	2.492	25	0.413	0.559	0.342	0.498
26	2.060	2.787	1.708	2.485	26	0.404	0.547	0.335	0.487
27	2.056	2.779	1.706	2.479	27	0.396	0.535	0.328	0.477
28	2.052	2.771	1.703	2.473	28	0.388	0.524	0.322	0.467
29	2.048	2.763	1.701	2.467	29	0.380	0.513	0.316	0.658
30	2.045	2.756	1.699	2.462	30	0.373	0.503	0.310	0.449
40	2.024	2.707	1.682	2.430	40	0.320	0.428	0.266	0.384
50	2.008	2.680	1.676	2.404	50	0.284	0.379	0.237	0.340
60	2.000	2.664	1.673	2.393	60	0.258	0.344	0.216	0.309

**Table 2 polymers-15-02757-t002:** Tensile test results of composite test samples.

Material	Extrusion Type	Sample	a_0_	b_0_	L_0_	Fm	σ_m_	ε_m_	Extrusion Type	Sample	a_0_	b_0_	L_0_	Fm	σ_m_	ε_m_	Extrusion Type	Sample	a_0_	b_0_	L_0_	Fm	σ_m_	ε_m_
		Designation	[mm]	[mm]	[mm]	[N]	[MPa]	[%]		Designation	[mm]	[mm]	[mm]	[N]	[MPa]	[%]		Designation	[mm]	[mm]	[mm]	[N]	[MPa]	[%]
Carbon PLA	Length	11A	4.04	10.57	50.00	995	23	1.60		12A	4.02	10.13	50.00	46	1	0.36		13A	3.88	10.19	50.00	573	15	2.32
		11B	3.80	11.47	50.00	877	20	2.15		12B	4.00	10.03	50.00	47	1	0.24		13B	4.01	9.97	50.00	536	13	3.66
		11C	4.06	10.58	50.00	921	21	1.97		12C	3.94	10.06	50.00	74	2	0.22		13C	4.04	10.11	50.00	536	13	3.01
		11D	3.87	10.64	50.00	886	22	2.06		12D	3.92	10.26	50.00	53	1	0.13		13D	3.98	10.10	50.00	616	15	2.37
		11E	4.08	10.91	50.00	886	20	1.97		12E	3.98	9.93	50.00	58	1	0.15		13E	3.92	10.05	50.00	586	15	2.52
		11F	3.95	10.95	50.00	905	21	2.08	Height	12F	3.92	10.03	50.00	19	1	0.10	Width	13F	4.05	9.94	50.00	555	14	2.96
		11G	3.85	10.88	50.00	846	20	1.77		12G	4.00	9.93	50.00	44	1	0.14		13G	3.97	9.90	50.00	578	15	3.31
		11H	3.83	10.22	50.00	891	23	2.06		12H	4.02	10.08	50.00	132	3	0.54		13H	4.01	9.92	50.00	617	16	2.43
		11I	3.87	10.42	50.00	866	21	1.98		12I	4.07	10.02	50.00	34	1	0.29		13I	3.99	9.96	50.00	565	14	3.49
		11J	3.96	10.75	50.00	885	21	2.01		12J	4.00	10.09	50.00	103	3	0.34		13J	4.03	10.04	50.00	544	13	3.28
Carbon PLA white	Length	14-1A	4.50	10.07	50.00	1051	23	2.25		14-2A	4.12	10.05	50.00	56	1	0.07		14-3A	4.05	9.96	50.00	818	20	2.04
		14-1B	4.02	10.14	50.00	1091	27	2.14		14-2B	3.87	9.93	50.00	37	1	0.31		14-3B	4.04	10.01	50.00	831	21	2.45
		14-1C	4.01	10.06	50.00	1061	26	2.06		14-2C	4.02	10.05	50.00	42	1	0.02		14-3C	4.03	10.04	50.00	793	20	1.93
		14-1D	3.90	10.19	50.00	1055	27	1.64		14-2D	4.15	10.09	50.00	55	1	0.03		14-3D	4.05	10.10	50.00	802	20	1.87
		14-1E	3.95	10.25	50.00	1068	26	2.10		14-2E	4.01	9.89	50.00	28	1	0.15	Width	14-3E	4.03	9.09	50.00	845	23	2.19
		14-1F	3.98	10.14	50.00	1053	26	2.12	Height	14-2F	4.05	9.92	50.00	11	1	0.24		14-3F	4.02	9.92	50.00	815	20	2.19
		14-1G	3.90	10.06	50.00	1039	26	2.26		14-2G	4.18	10.04	50.00	157	4	0.81		14-3G	4.02	9.89	50.00	865	22	2.23
		14-1H	3.82	10.34	50.00	1037	26	2.11		14-2H	3.94	10.08	50.00	82	2	0.74		14-3H	3.95	9.97	50.00	788	20	2.02
		14-1I	4.08	10.13	50.00	1069	26	2.20		14-2I	3.97	10.12	50.00	109	3	0.17		14-3I	4.03	9.93	50.00	821	21	2.14
		14-1J	4.02	10.08	50.00	1020	25	2.05		14-2J	4.01	9.89	50.00	28	1	0.15		14-3J	4.05	9.95	50.00	824	23	2.15
Bronze	Length	21A	3.96	10.62	50.00	673	16	1.26		22A	4.01	9.93	50.00	104	3	0.31		23A	4.01	9.98	50.00	805	2	1.94
		21B	4.02	10.50	50.00	724	17	1.35		22B	3.98	10.02	50.00	116	3	0.16		23B	4.01	10.07	50.00	804	20	2.12
		21C	4.01	10.53	50.00	524	12	0.95		22C	3.98	10.01	50.00	107	3	0.37		23C	4.00	10.12	50.00	785	19	1.93
		21D	4.01	10.86	50.00	709	16	1.28		22D	4.07	9.93	50.00	108	3	0.23		23D	4.01	10.07	50.00	804	20	1.95
		21E	4.03	10.76	50.00	734	17	1.40		22E	4.00	10.03	50.00	102	3	0.50		23E	3.99	10.48	50.00	866	21	2.25
		21F	3.99	10.86	50.00	567	13	1.10	Height	22F	4.20	10.04	50.00	94	2	0.28	Width	23F	4.06	10.04	50.00	778	19	1.89
		21G	3.99	10.46	50.00	850	20	1.56		22G	3.98	10.02	50.00	116	2	0.16		23G	3.99	10.25	50.00	817	20	1.76
		21H	3.96	10.44	50.00	947	23	1.74		22H	4.07	9.93	50.00	108	3	0.23		23H	3.99	10.04	50.00	829	21	1.96
		21I	3.89	10.67	50.00	726	17	1.41		22I	4.00	10.03	50.00	102	2	0.50		23I	4.00	10.34	50.00	809	20	1.81
		21J	4.01	10.08	50.00	745	18	1.42		22J	3.98	10.02	50.00	116	3	0.16		23J	3.95	10.26	50.00	887	22	2.19
Bronze PLA Carbon	Length	21-1A	4.27	10.45	50.00	979	22	2.13		21-2A	4.26	10.25	50.00	73	2	0.30		21-3A	4.02	10.26	50.00	757	18	2.38
		21-1B	4.07	10.25	50.00	989	24	2.12		21-2B	3.97	9.06	50.00	62	2	0.36		21-3B	3.95	10.28	50.00	812	20	2.24
		21-1C	4.01	10.66	50.00	966	22	2.14		21-2C	4.17	10.09	50.00	71	2	0.05		21-3C	4.01	10.90	50.00	801	18	2.21
		21-1D	4.01	10.29	50.00	986	24	2.16		21-2D	3.96	10.10	50.00	70	2	0.49		21-3D	4.02	10.10	50.00	806	20	2.31
		21-1E	4.01	10.24	50.00	950	23	2.09		21-2E	4.05	10.19	50.00	64	2	0.58		21-3E	3.97	10.90	50.00	761	18	2.40
		21-1F	4.07	10.56	50.00	983	23	2.21	Height	21-2F	3.97	9.06	50.00	62	2	0.39	Width	21-3F	4.04	10.04	50.00	782	19	2.13
		21-1G	4.11	10.20	50.00	984	23	2.13		21-2G	3.96	10.20	50.00	70	2	0.49		21-3G	4.00	10.32	50.00	776	19	2.38
		21-1H	4.10	10.26	50.00	920	22	2.01		21-2H	3.98	9.06	50.00	63	2	0.38		21-3H	3.97	10.25	50.00	787	19	2.45
		21-1I	4.03	10.26	50.00	966	23	2.13		21-2I	4.05	10.19	50.00	64	2	0.58		21-3I	3.99	10.14	50.00	800	20	2.23
		21-1J	4.12	10.79	50.00	956	21	1.94		21-2J	4.18	10.29	50.00	71	2	0.25		21-3J	4.02	10.12	50.00	720	18	1.99
Bronze PLA White	Length	24-1A	4.16	10.17	50.00	1076	25	1.85		24-2A	4.16	10.02	50.00	62	1	0.06		24-3A	4.01	10.10	50.00	658	16	1.71
		24-1B	4.12	10.20	50.00	1134	27	2.07		24-2B	4.14	10.15	50.00	61	1	0.12		24-3B	4.03	10.24	50.00	960	23	2.27
		24-1C	4.27	10.20	50.00	1140	26	2.17		24-2C	4.38	10.30	50.00	68	2	0.41		24-3C	4.00	10.21	50.00	840	21	2.47
		24-1D	4.23	1061	50.00	925	21	1.55		24-2D	4.23	9.98	50.00	61	1	0.39		24-3D	3.99	10.04	50.00	784	20	2.20
		24-1E	4.08	10.48	50.00	1138	27	2.05		24-2E	4.09	10.06	50.00	60	1	0.75		24-3E	4.07	10.08	50.00	781	19	2.12
		24-1F	4.13	10.46	50.00	1121	26	2.00	Height	24-2F	4.10	10.05	50.00	62	1	0.06	Width	24-3F	4.00	10.10	50.00	823	20	2.24
		24-1G	4.07	10.15	50.00	1103	27	1.95		24-2G	4.15	10.17	50.00	61	1	0.13		24-3G	3.99	10.16	50.00	824	20	2.33
		24-1H	4.12	10.20	50.00	1176	28	2.06		24-2H	4.39	11.15	50.00	68	2	0.43		24-3H	4.07	10.14	50.00	751	18	2.32
		24-1I	4.15	10.14	50.00	1020	24	1.76		24-2I	4.25	9.99	50.00	61	1	0.41		24-3I	3.97	10.27	50.00	730	18	2.08
		24-1J	4.10	10.67	50.00	868	20	1.58		24-2J	4.09	10.07	50.00	60	1	0.75		24-3J	4.06	10.05	50.00	706	17	2.08

## Data Availability

Not applicable.

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
