# Peer review of "Investigation of the Manufacturing Orientation Impact on the Mechanical Properties of Composite Fiber-Reinforced Polymer Elements in the Fused Filament Fabrication Process"

_polymers, 2023, doi:10.3390/polym15132757_

Round 1

Reviewer 1 Report

The article is facing major challenges in different sections and the most basic issues such as the organization of the article, numbering of sections and subsections, the manner and order of referencing, presentation of images, analysis of results have not been observed. Some of these deficiencies are as follows. Also, the current version lacks research value.

The way of referencing the articles is very awful and disappointing. The use of general sentences with more than 5 references can be seen in all parts of the introduction. On the other hand, appropriate references were not used in the analysis of the results. References should also be numbered in order (Lines 37 and 65 and 208).

“Measured force values Fm is force at the strength limit, which is the highest load repre- 205 senting the maximum values of the measured force obtained during the test. From the 206 given force value, it is possible to determine the tensile strength limit σM and the propor- 207 tional elongation at tensile strength εM (Table 1-5) [20-32].”

Almost half of the references in the article are from the authors themselves, and this is not acceptable, especially the use of conference resources that are about 10 years ago. This field is quite up-to-date and new sources should be used. References No. 12, 14, 17-19 and 27-29 should be deleted.

The first paragraph of the introduction (lines 32 to 41) should be deleted. The second paragraph of the introduction should also be deleted. The items mentioned in this section are very old and general.

The numbering of sections and subsections is messed up (Line 98).

The number of images used are very large and some of them are not of the required quality and should be removed. Also, some should be presented in the form of a table.

Figure one should be deleted. This image lacks quality. Also, the schematic of the FDM process is presented in Figure 2.

Tables 1 to 5 should be presented as a stress-strain diagram. Diagrams 6 to 10 are screenshots. Figures 14 to 23, vertical and horizontal axes are not defined for them. All images are scale-free.

No comment.

Author Response

The article is facing major challenges in different sections and the most basic issues such as the organization of the article, numbering of sections and subsections, the manner and order of referencing, presentation of images, analysis of results have not been observed. Some of these deficiencies are as follows. Also, the current version lacks research value.

Revised manuscript has corrected numbering of sections and subsections, references, literature review was improved. Presentation of results using images and graphs was improved. Analysis of the results was introduced into discussion section of the manuscript. In our opinion, revised manuscript was improved in such a way that it does not lack the research value.

The way of referencing the articles is very awful and disappointing. The use of general sentences with more than 5 references can be seen in all parts of the introduction. On the other hand, appropriate references were not used in the analysis of the results. References should also be numbered in order (Lines 37 and 65 and 208).

More in depth review of current research regarding 3D printing of composite materials was implemented in the introduction section of this paper. Also, references were put in order.

“Measured force values Fm is force at the strength limit, which is the highest load repre- 205 senting the maximum values of the measured force obtained during the test. From the 206 given force value, it is possible to determine the tensile strength limit σM and the propor- 207 tional elongation at tensile strength εM (Table 1-5) [20-32].”

References in this paragraph were erased. Also, text in this paragraph was improved.

Almost half of the references in the article are from the authors themselves, and this is not acceptable, especially the use of conference resources that are about 10 years ago. This field is quite up-to-date and new sources should be used. References No. 12, 14, 17-19 and 27-29 should be deleted.

Most of the references that included authors were removed from manuscript. New sources were included in introduction section of the paper. References added in this section are from recent research works.

The first paragraph of the introduction (lines 32 to 41) should be deleted. The second paragraph of the introduction should also be deleted. The items mentioned in this section are very old and general.

The paragraph of the introduction (lines 32 to 41) was revised. The second paragraph was deleted.

The numbering of sections and subsections is messed up (Line 98).

Numbering of sections and subsections was corrected in the revised version of the manuscript.

The number of images used are very large and some of them are not of the required quality and should be removed. Also, some should be presented in the form of a table.

The number of images was reduced. The quality of existing images was improved.

Figure one should be deleted. This image lacks quality. Also, the schematic of the FDM process is presented in Figure 2.

Figure one in original version of manuscript was removed.

 Tables 1 to 5 should be presented as a stress-strain diagram. Diagrams 6 to 10 are screenshots. Figures 14 to 23, vertical and horizontal axes are not defined for them. All images are scale-free.

Table 1 to 5 were combined into one table according to academic editor suggestion. Diagrams 6 to 10 were removed. Instead, diagrams from tensile test were introduced in the revised paper. Figures 14 to 23 were improved; vertical and horizontal axes are now defined.

Reviewer 2 Report

The authors conducted tensile tests to study the mechanical properties of selected composite materials, considering their length, height, and width orientations, and analyzed the corresponding results. I recommend the publication of the paper if the authors address the following comments:

1. The resolution of some figures, including Figs. 3-5 and 6-10, needs improvement. The texts and x and y axes in Figs. 6-10 are difficult to read.

2. The quality of English writing needs improvement, as some sentences in the article are difficult to understand. For instance, “This proves which manufactured composite samples are in which direction and with which PLA composite materials are the strongest.”

The quality of English writing needs improvement.

Author Response

The authors conducted tensile tests to study the mechanical properties of selected composite materials, considering their length, height, and width orientations, and analyzed the corresponding results. I recommend the publication of the paper if the authors address the following comments:

  1. The resolution of some figures, including Figs. 3-5 and 6-10, needs improvement. The texts and x and y axes in Figs. 6-10 are difficult to read.

Most of the figures in the manuscript were improved. New diagrams representing deformation behavior of tested materials were introduced, they replaced Fig. 6-10.

  1. The quality of English writing needs improvement, as some sentences in the article are difficult to understand. For instance, “This proves which manufactured composite samples are in which direction and with which PLA composite materials are the strongest.”

The quality of English writing was improved, thus sentences in the article should be easier to understand.

Author Response

The manuscript entitled: Investigation of the Best Manufacturing Orientation of Composite Fibre- reinforced Polymers Elements in the Fused Filament Fabrication Process presents interesting research results.

This paper analyses the mechanical properties of selected PLA-based composite materials - PolyLacticAcid - and its composite variants. The analysis of the experimental data showed thecorrelation between the choice of orientation and the layering strategy of the selected filaments and how much the print orientation and the combination of different filaments influence the strength, ductility.

The research results described in the article correspond to the topics of the Polymers. The following comments will help to improve the manuscript:

1) The paper presents experimental tests on tensile specimens according to EN 176 ISO 527 and 2602. However, a description and presentation of the detailed test rig is missing. The necessary figures and description should be added.

Description of test rig, test samples was implemented into the article. The necessary figures and description were added in the revised version of paper.

2) The composite is composed in a matrix and reinforcement. As part of the 3D printing, was only the reinforcement made and the matrix missed?

The PLA material was always used as matrix in test samples. Different types of reinforcement were used in tensile tests, two types of PLA filaments were used in experiments.

3) It is worth adding figures and a description of the workstation for macroscopic measurements (Keyence VHX 237 - 500 digital microscope).

The figure and description of digital microscopic was added to the revised paper.

4) Due to the topic of the paper and the type of analysis, it is worth extending the introduction to the topic. Manuscripts presenting experimental studies of layered fiber polymer composites may be

helpful: 10.1063/1.5092009; 10.3390/ma14061506

The introduction was extended by two manuscripts you mentioned.